# Predominant Polarity and Polarity Index of Maintenance Treatments for Bipolar Disorder: A Validation Study in a Large Naturalistic Sample in Italy

**DOI:** 10.3390/medicina57060598

**Published:** 2021-06-10

**Authors:** Umberto Albert, Mirko Manchia, Sofia Burato, Bernardo Carpiniello, Gabriele Di Salvo, Federica Pinna, Gianluca Rosso, Giuseppe Maina

**Affiliations:** 1Department of Medicine, Surgery and Health Sciences, University of Trieste, 34128 Trieste, Italy; 2Department of Mental Health, Psychiatric Clinic, Azienda Sanitaria Universitaria Giuliano-Isontina—ASUGI, 34128 Trieste, Italy; sofia.burato@gmail.com; 3Section of Psychiatry, Department of Medical Sciences and Public Health, University of Cagliari, 09127 Cagliari, Italy; mirkomanchia@unica.it (M.M.); bcarpiniello@unica.it (B.C.); fedepinna@inwind.it (F.P.); 4Unit of Clinical Psychiatry, University Hospital Agency of Cagliari, 09127 Cagliari, Italy; 5Department of Pharmacology, Dalhousie University, Halifax, NS B3H4R2, Canada; 6Department of Neurosciences Rita Levi Montalcini, University of Turin, 10100 Turin, Italy; gabriele.disalvo@unito.it (G.D.S.); gianluca.rosso@unito.it (G.R.); giuseppe.maina@unito.it (G.M.); 7Psychiatric Unit, San Luigi Gonzaga University Hospital, 10100 Turin, Italy

**Keywords:** bipolar disorder, predominant polarity, polarity index

## Abstract

*Background and Objectives*: Predominant polarity (PP) may be a useful course specifier in at least a significant proportion of patients with Bipolar Disorder (BD), being associated with several clinically relevant correlates. Emerging evidence suggests that the concept of PP might influence the selection of maintenance treatments, based on a drug polarity index (PI) which measures the greater antidepressive vs. antimanic preventive efficacy of mood stabilizers over long-term maintenance treatment. In this study, we aimed to validate the PI in a large sample of Italian BD patients with accurate longitudinal characterization of the clinical course, which ensured a robust definition of the PP. *Materials and Methods*: Our sample is comprised of 653 patients with BD, divided into groups based on the predominant polarity (manic/hypomanic predominant polarity—MPP, depressive predominant polarity—DPP and no predominant polarity). Subsequently we calculated the mean total polarity index for each group, and we compared the groups. *Results*: When we examined the mean PI of treatments prescribed to individuals with DPP, MPP and no predominant polarity, calculated using two different methods, we failed to find significant differences, with the exception of the PI calculated with the Popovic method and using the less stringent criterion for predominant polarity (PP_50%_). *Conclusions*: Future prospective studies are needed in order to determine whether the predominant polarity is indeed one clinical factor that might guide the clinician in choosing the right mood stabilizer for BD maintenance treatment.

## 1. Introduction

Bipolar disorder (BD) is a highly heterogeneous psychiatric condition in terms of phenotypic expression. This relatively high degree of heterogeneity hinders, on the one hand, the formulation of an accurate diagnosis, which in turn might result in significant delays in prescribing an adequate treatment [1]. On the other, it might lead to a difficulty in selecting the right treatment for the right patient at the right time [2].

The selection of the most adequate treatment, both in terms of effectiveness and safety, is particularly relevant in the very long-term, even a lifelong, perspective that characterizes the longitudinal trajectory of BD. This step relies extensively on the clinical delineation at the individual level of patients affected by BD: the information conveyed by the diagnosis, in fact, is in itself insufficient for therapeutic and prognostic purposes [3]. Some examples of salient domains that should be considered for the clinical characterization of the adult patient with a major depressive disorder [3] or primary psychosis [4] have been recently described. Concerning BD, the need of a personalized (tailored) management of the disorder has been emphasized, both for a better comprehension of the neurobiological underpinnings of BD as well as for the development of treatments for targeted subgroups of patients with the final aim of improving response rates [5,6,7,8].

Among several clinical factors (e.g., diagnostic subtypes, age at onset, presence of lifetime comorbid disorders, etc.) that may help clinicians to personalize the management of BD [9,10,11,12]), course specifiers such as the cycle pattern (Mania-Depression-Interval—MDI—vs. Depression-Mania-Interval—DMI—course) [13,14,15,16,17] or the predominant polarity have been proposed and validated [18,19,20].

Angst [18] initially proposed the concept of predominant polarity based on his long-term naturalistic follow-up of patients with manic-depressive illness. Subsequently, Colom and colleagues [19] operationalized this clinical construct and defined it as one polarity occurring during at least two-thirds of lifetime episodes and identified two groups: depressive predominant polarity (DPP) for BD patients experiencing at least two-thirds of lifetime depressive episodes and manic/hypomanic predominant polarity (MPP) for those with at least two-thirds of lifetime manic/hypomanic episodes. This operationalized definition has been recommended by the International Society for Bipolar Disorder Task Force for the nomenclature of course and outcome in bipolar disorders [20].

An alternative threshold has been used by a minority of authors, which considered patients to have a predominant polarity when the total number of episodes of one pole (e.g., depressive) exceeded the total number of episodes of the opposite pole (≥51% excess of one polarity) [21,22,23,24]. Two studies [25,26] used both definitions of predominant polarity.

Approximately half of all patients with BD can be categorized according to a predominant polarity (2/3 definition of predominant polarity), with, however, a wide variation between different studies (28.9% to 64%) [20,27]. This wide variation may be partly explained by differences in the samples included in each study, for example in the proportion of patients with BD type I vs. type II. Moreover, another contributor to the wide variation found between different studies may be the definition of bipolar disorder itself; it is possible that different approaches to the diagnosis of BD (e.g., tendencies to overdiagnose personality disorders instead of milder forms of BD, or failing to recognize hypomanic episodes in the history of patients with recurring depressive episodes, thus resulting in a diagnosis of recurrent MDD instead of BD type II) could have contributed to selecting different groups of individuals with BD in each study.

Among those with a predominant polarity, DPP appears to be more prevalent in the majority of the studies (DPP: 17–50% of all bipolar patients—with or without a predominant polarity—vs. 12–38% for MPP) [19,26,27,28,29,30,31,32,33,34,35,36,37].

Predominant polarity may be a useful course specifier in at least a significant proportion of patients with BD, it being associated with several clinically relevant correlates. DPP is associated with female gender, being married, BD type II, a depressive polarity at onset, more lifetime mood episodes, higher suicide risk, and a higher number of years untreated, while MPP is associated with male gender, BD type I, younger age at onset, (hypo)manic polarity of the first episode lifetime, greater comorbidity with substance use disorders, more hospitalizations, and higher prevalence of psychotic symptoms [20,27,34]. Cognitive impairment was found to be more frequent in MPP [38].

Emerging evidence suggests that the concept of predominant polarity (course specifier) may influence the selection of maintenance treatments.

Some mood stabilizers have a greater effect in protecting from depressive or (hypo)manic recurrences over long-term maintenance treatment. Such relative antidepressive vs. antimanic preventive efficacy of drugs may be measured, can be considered an attribute of the specific pharmacological compound, and is termed polarity index (PI) [39]. Polarity index was firstly retrieved by Popovic and colleagues (2012); they systematically reviewed randomized controlled trials of medications used for the maintenance treatment of BD (mood stabilizers or antipsychotic drugs alone or in combination with a mood stabilizing agent such as lithium or valproate vs. placebo). The Number Needed to Treat (NNT) was calculated by taking the reciprocals of the differences between the rates of the outcomes for two interventions; the polarity index was then retrieved by dividing NNT for prevention of depressive episodes and NNT for the prevention of manic episodes. A PI value above 1.0 indicates a relatively greater antimanic prophylactic efficacy while number below 1.0 a relative greater antidepressive efficacy [39]. According to this method, antipsychotics with mood stabilizing properties and classic mood stabilizers may be ranked in accordance to their prevalent prophylactic efficacy. However, the assignment of a specific compound to the antimanic or antidepressant mood stabilizing end is controversial, depending on the different number of studies included in the calculation of the PI [39,40].

In recent years, few authors investigated whether the choice of the right mood stabilizer over the long-term in terms of PI is influenced by the predominant polarity as a course specifier of the disorder (which is whether clinicians prescribe more often to individuals with a manic predominant polarity a mood stabilizer with greater effectiveness in preventing manic episodes over depressive ones and thus with a higher PI). No prospective, longitudinal studies exist examining the relative effectiveness of a mood stabilizer chosen for its polarity index according to a clinical characterization based on the predominant polarity. We have only a few retrospective studies which have evaluated cross-sectionally whether subjects with MPP received mood stabilizers with a higher PI (that is, with a greater effectiveness in preventing manic episodes over depressive ones) as compared to subjects with DPP.

The results are controversial. Popovic et al. [36], in a sample of 604 individuals found that total PI, antipsychotics’ PI and mood stabilizers’ PI were all significantly higher in individuals with MPP, confirming the usefulness of the PI construct. In this study, they calculated the mean PI as from Popovic et al. (2012) [39]. A different validation study that used the same calculation of the PI in a German sample [37] failed to find differences between subjects with MPP and those with DPP in total PI and antipsychotics’ PI, with only the PI of mood stabilizers lower in patients with DPP, indicating a stronger antidepressant regimen in the DPP group. A limitation of these studies is that they did not include valproate and carbamazepine in their analyses, as a PI to carbamazepine was not assigned by Popovic and colleagues [39] due to the lack of a long-term placebo-controlled trial, and the PI for valproate was considered unreliable because the pivotal trial did not show statistical superiority of valproate over placebo. Moreover, PI for newer agents such as paliperidone or asenapine were lacking. A third study, although not focused on the validation of the PI, failed to find any differences in mean PI values (total, antipsychotics and mood stabilizers) between predominantly manic and predominantly depressed groups [27].

There is a strong need for new studies aimed at validating the PI in large samples from the real world. Moreover, Carvalho et al. [40] recalculated polarity indexes of maintenance treatments used for BD, including new randomized controlled trials, leading to some differences in the PI of some compounds as compared to those from Popovic et al. [39] It is therefore possible that some differences might emerge using different PI calculations.

### Aims

We aimed to validate the polarity index (PI) in a large sample of Italian BD patients with accurate longitudinal characterization of the clinical course, which ensured a robust definition of the PP. In agreement with previous evidence, we hypothesized that the presence of drug treatments with PI > 1.0 would associate with BD patients with manic PP (MPP), while those with PI < 1.0 to BD patients with depressive PP (DPP).

## 2. Materials and Methods

### 2.1. Clinical Sample

Our sample is comprised of 653 patients with BD diagnosed according to the Diagnostic and Statistical Manual of Mental Disorders, Fifth Edition [41]. Specifically, 262 patients had BD type I (BDI), 371 had BD type II (BDII), 19 had unspecified BD, and 1 had substance/medication-induced BD.

Clinical records of inpatients and outpatients with a diagnosis of BD consecutively admitted or referring to the Psychiatric Units of the San Luigi Gonzaga Hospital in Orbassano and the Molinette Hospital in Torino (University of Turin, Italy) were analyzed for the present study. All subjects had given a written informed consent to have their clinical data potentially used for research purposes (provided that these data are anonymously treated). The present analysis is part of an independent retrospective observational study on the clinical characterization of BD which has been reviewed and approved by the local Ethical Committee (Prot. 7119, 18 April 2018, Comitato Etico Interaziendale A.O.U. San Luigi Gonzaga di Orbassano AA.SS.LL. TO3-TO4-TO5).

Certified psychiatrists with at least four years of postgraduate clinical experience performed the clinical assessment of patients. All potential interviewers met prior to the study beginning and underwent common extensive training prior to conducting the assessments. A systematic review of patients’ medical records helped clinicians to corroborate data concerning the clinical characteristics of the disorder emerging from the direct interview, particularly those related to the longitudinal illness course. Clinical data were used to depict the longitudinal course of the illness with the life chart method [42]. This method allows the identification of both depressive and manic polarities, and rates the severity and duration of episodes, as well as characterizing course sequences with major or minor depressive (D or d) alternating (preceding or following) manic, mixed, or hypomanic episodes (M, Mx, or m) and euthymic intervals (I). External corroboration for clinical data was obtained, whenever possible, by directly interviewing, with the patient’s consent, a first-degree family member or other significant individuals.

The retrospective examination of clinical charts of patients was carried out from June 2018 to December 2019. Statistical analyses concerning the present study were performed in December 2020.

### 2.2. Assessment of Polarity Index and Predominant Polarity

The polarity index was calculated for each patient according to the method described in Popovic et al. (2012) [39] (Polarity Index Popovic) and that in Carvalho et al. (2015) [40] (Polarity Index Carvalho). Popovic and colleagues [39] included 16 maintenance trials, while Carvalho et al. [40] included 18. NNT was calculated by taking the reciprocals of the differences between the rates of the outcomes between the interventions; the PI was retrieved by dividing the NNT for prevention of depressive episodes by the NNT for the prevention of manic ones.

According to Popovic at al. [39], the PI values for each drug were as follows: 12.09 for risperidone LAI, 4.38 for aripiprazole, 3.91 for ziprasidone, 2.98 for olanzapine, 1.39 for lithium, 1.14 for quetiapine, 0.62 for oxcarbazepine, 0.49 for valproate and 0.40 for lamotrigine. Carvalho and colleagues [40] recalculated, on the basis of different and new trials, the polarity indexes of maintenance drugs for BD and provided the following PI for each compound: 10.4 for aripiprazole monotherapy, 9.1 for risperidone LAI, 4.2 for aripiprazole adjunctive to lithium/divalproex, 4.0 for olanzapine monotherapy, 3.9 for ziprasidone adjunctive to lithium/divalproex, 1.4 for lithium and for quetiapine monotherapy, 0.8 for quetiapine combined with lithium/divalproex, 0.6 for oxcarbazepine combined with lithium, 0.5 for olanzapine combined with lithium/divalproex and for divalproex, and 0.4 for lamotrigine.

Although a PI was assigned to valproate and oxcarbazepine, it was considered unreliable because the pivotal trials did not show the statistical superiority of those two drugs over placebo. We then excluded from the analysis patients prescribed drugs without an assigned PI (e.g., patients on valproate or oxcarbazepine, or on antipsychotic such as paliperidone).

Specifically, PI was calculated for the current treatment of each patient, independently from the prescribed dosage; when patients received more than one pharmacological treatment, PI was calculated as a mean of all the prescribed treatments as in Popovic et al. (2014). Subsequently, the mean total polarity index for each group (MPP, DPP and no predominant polarity) was calculated, and the groups were compared.

Predominant polarity was defined as at least half (PP_50%_) or two-thirds (PP_2/3_) of a patient’s past episodes fulfilling DSM-5 criteria for major depressive episode or manic or hypomanic episodes [19,25]. By default, the PP_50%_ group would be subsumed into the PP_2/3_ group. We chose to consider also the 50% predominant polarity criterion as some authors previously suggested to be less stringent in defining this course specifier, which is easier to use in clinical practice [21,22,23,24,25,26].

### 2.3. Statistical Analysis

We compared the main clinical characteristics of BD patients with MPP and DPP using univariate analysis (*t*/Mann–Whitney test or *Χ*^2^ test, as appropriate). When one or more cells had expected values of 5 or less, Fisher’s exact test was used in 2 × 2 contingency tables and bootstrap with 1000 samples in larger tables. Only clinical variables presenting a statistically significant association with a PP subgroup (*p* < 0.05) were entered into a backward stepwise multivariate binary logistic model to account for possible intercorrelations. A polarity index was compared between BD patients with MPP, DPP, and those with an absence of predominant polarity, by means of ANOVA or a Kruskal–Wallis (KW) test. Either of the latter tests were applied, as appropriate, depending on the presence of homoscedasticity, which was assessed with the Levene’s test. Statistical significance was set at α = 0.05. All statistical analyses were performed with IBM^®^ SPSS^®^ Statistics, Version 24.

## 3. Results

### 3.1. Clinical Sample

Socio-demographic and clinical characteristics of the sample are summarized in Table 1. All subjects were of Italian ancestry. Approximately 60% of the total sample consisted of women. The mean age at interview and age at onset were at 50.6 years (standard deviation [SD] ± 15.6) and 30.3 years (SD ± 12.9), respectively. The mean illness duration was 20.2 years (SD ± 13.9). These findings are summarized in Table 1.

### 3.2. Clinical Characteristics of Bipolar Disorder Patients According to PP_2/3_ or PP_50%_

Two-hundred and thirty-six (36.1%) BD patients had a DPP_2/3_, 70 (10.7%) had a MPP_2/3_, while 347 (53.1%) BD patients did not meet criteria for either PP_2/3_. Concerning PP_50%_, we found that 118 (18.1%) BD patients had a MPP_50%_, 361 (55.3%) had a DPP_50%_, while 174 (26.6%) had absence of PP_50%_ (Table 1).

The comparison of clinical characteristics between MPP_2/3_ and DPP_2/3_ identified differences in terms of age at interview and the presence of lifetime suicidal behavior (Table 2).

Specifically, MPP_2/3_ BD patients had a lower age at interview compared to DPP ones (*t* = −4.1, *p* < 0.0001). Furthermore, DPP_2/3_ BD patients had a higher rate of lifetime suicidal behavior compared to MPP_2/3_ ones (χ^2^ = 10.1, *p* = 0.001). As expected, MPP_2/3_ BD patients had a higher number of manic episodes (U = 3536.0, *p* < 0.0001), and, conversely, DPP_2/3_ BD patients had a higher number of depressive episodes (U = 1413.0, *p* < 0.0001). A logistic regression model confirmed that lifetime suicidal behavior and age at interview was associated with DPP_2/3_ (Table 3). As there was a significant correlation between the number of manic and of depressive episodes with the respective PP categories, we did not include these variables in the logistic regression model.

When we applied the PP_50%_ criterion, the comparison between MPP_50%_ and DPP_50%_ identified differences in terms of age at interview, illness duration, gender, presence of lifetime suicidal behavior, type of clinical course cycle, and the number of manic and depressive episodes. Specifically, compared to MPP_50%_, DPP_50%_ BD patients were more likely to be females (χ^2^ = 6.2, *p* = 0.016) and had an older age at interview (*t* = −4.1, *p* < 0.0001) as well as a higher rate of lifetime suicidal behavior (χ^2^ = 4.5, *p* = 0.034) and a longer duration of illness (*t* = −3.6, *p* < 0.0001). Furthermore, they showed a higher rate of DMI clinical course compared to MPP_50%_ (16.1% vs. 7.6%, χ^2^ = 11.9, *p* = 0.018). Conversely, MPP_50%_ showed a higher number of manic (U = 9882.0, *p* < 0.0001) and depressive (U = 7898.0, *p* < 0.0001) episodes (Table 2).

A logistic regression model showed that only lifetime suicidal behavior and age at interview were significantly associated with DPP_50%_ after correcting for intercorrelations (Table 3). Again, due to the significant correlation between the number of manic and of depressive episodes with the MPP_50%_ subgroup, we did not include these variables in the logistic regression model.

### 3.3. Comparison of Polarity Index among PP_2/3_ or PP_50%_ Bipolar Disorders Subgroups

Analysis of variance found a difference between PP_50%_ subgroups in values of total PI defined according to Popovic et al. 2012 (Table 4). Specifically, the mean total PI was significantly higher in MPP_50%_ (1.91 ± 0.77) compared to DPP_50%_ (1.54 ± 0.76) and to BD patients with no PP_50%_ (1.61 ± 0.57) (F = 4.038, *p* = 0.018). Since Levene’s test found that the variance of PI defined according to Carvalho et al. 2015 was not homogenous among PP_50%_ BD subgroups, we applied the KW test without identifying a statistically significant difference. All other comparisons did not find statistically significant differences in mean PI values among PP_2/3_ or PP_50%_ BD subgroups (summarized in Table 4).

When we ran the analysis on the subset of patients with BD type I we did not find other significant differences.

## 4. Discussion

Predominant polarity has been proposed as a useful clinical course specifier of BD, being associated with several clinically relevant correlates such as BD subtype, polarity of the first episode, a higher suicide risk, number of mood episodes and hospitalizations, lifetime psychotic symptoms, and comorbidity with substance use disorders [7,34], which may prove to be useful in the personalized management of BD. However, this course specifier would become even more relevant in the clinical characterization of the disorder if one could demonstrate that it predicts a greater efficacy of prophylactic compounds with different PIs, chosen according to the predominant polarity (DPP associated with greater effectiveness of mood stabilizers with PI < 1.0; MPP associated with greater effectiveness of compounds with PI > 1.0). No prospective longitudinal studies exist, to our knowledge, examining the relative effectiveness of mood stabilizers prescribed over the long-term according to the predominant polarity.

Thus, an indirect way of examining the issue is to examine in the real-world whether individuals with BD are receiving, over the long-term, mood stabilizers with different PIs according to the predominant polarity. The few studies that tried to validate the PI found controversial results: Popovic et al. [36] confirmed the usefulness of the PI construct (total, antipsychotics and mood stabilizer PIs are significantly lower in individuals with DPP), while Volkert et al. [37] and Sentissi et al. [27] failed to replicate these findings (in the Volkert et al. study only the PI of mood stabilizers was significantly lower in individuals with DPP). Among the possible reasons for these discrepancies are the limited sample size of some studies, the lack of data on PI of valproate and carbamazepine, or the different settings/prescription patterns. It is also possible that psychiatrists in the Barcelona Bipolar Disorder Program are more sensitive to the PP concept when choosing a maintenance treatment for their patients.

In the present study, we aimed to validate the PI in a large sample of Italian BD patients with an accurate longitudinal characterization of the clinical course, which ensured a robust definition of the PP. In agreement with previous evidence, we hypothesized that the presence of drug treatments with PI > 1.0 would be associated with BD patients with manic PP (MPP), while those with PI <1.0 to BD patients with depressive PP (DPP).

In our sample using the recommended definition of PP (at least 2/3 of lifetime episodes being of one polarity or the other) [20], we confirmed that approximately half of all patients may be assigned to a PP, with DPP being the most frequent one (36.1% vs. 10.7% MPP). This latter finding may also be due to a slightly more frequent BDII diagnosis in our sample, as it has been suggested that DPP is more frequent among patients with BD type II [28,29,36,43], while MPP appears to be more frequent in samples of exclusively type I BD patients [29,36]. When we used a less stringent definition of predominant polarity (≥51% excess of one polarity), as previously done by other authors [25,26], the proportion of individuals with assigned PP increased to 73.4%, again with a preponderance of DPP over MPP (55.3% vs. 18.1%).

Although this was not the primary aim of our study, we found that DPP course was associated with clinically relevant differences as compared to individuals with MPP, confirming the usefulness of this course specifier [34]. One of the most relevant clinical issues is suicidal behavior: we confirmed that DPP is indeed associated with a greater suicide risk [19,23,30,44,45,46]. We also found that using a less stringent criterion for PP (≥51% excess of one polarity) may result in being more informative, although the logistic regression model showed again that only lifetime suicidal behavior and age at interview was significantly associated with DPP. In evaluating the suicidal risk of an individual with BD, then, the characterization of the clinical course in terms of PP may be highly informative for close monitoring.

When we examined the mean PI of treatments prescribed to individuals with DPP, MPP and no predominant polarity, calculated using the two different methods [39,40], we failed to find significant differences, with the exception of the PI calculated with the Popovic method and using the less stringent criterion for predominant polarity (PP_50%_). Overall, our results are in agreement with those of Volkert and colleagues [37] and Sentissi et al. [27] The lack of significant findings may have several possible explanations: the first one is that Italian psychiatrists may prescribe maintenance treatments without considering the predominant polarity of individuals with BD, and do not consider in the real-world the PI of a compound to be a reliable factor that may guide the choice of a single prophylactic treatment. Again, psychiatrists in Barcelona may be more sensitive to the PP and PI concepts when choosing a maintenance treatment for their patients being the PI proposed for the first time by the group of the Barcelona Bipolar Disorder Program. Another possible explanation for the lack of significant findings is that we did not control for the length of previous exposure to maintenance treatments; it is possible that people with more stable maintenance treatments (prophylactic treatment maintained stable for months/years) could show differences in the mean PI of their treatments according to the PP. Another issue to be considered is that we assigned a PI to each drug without considering whether it was used at an appropriate dosage.

Given that the methodology used has several limitations (the cross-sectional evaluation of mood stabilizers prescribed does not take into account clinical factors other than the PP that might have guided the choice of the compound for the individual patient, nor do we know or controlled for the length of exposure to the mood stabilizer prescribed), we cannot conclude that the PI construct is not useful in clinical practice.

Prospective, longitudinal studies could in the near future examine whether prescribing mood stabilizers with a higher PI (thus a greater efficacy in preventing manic episodes over depressive ones) to patients with a MPP course of the disorder (and vice versa prescribing compounds with lower PI to individuals with DPP) is associated with greater effectiveness in stabilizing patients. Several socio-demographic and clinical factors other than the PP contribute to the choice of the prescribed stabilizing drug, so that future studies will help clinicians to identify those factors which may be more informative and reliable in predicting a response to mood stabilizers over the long-term.

More randomized controlled trials on old, off-patent, but still widely used drugs such as valproate and carbamazepine are also needed, in order to help predict/refine their polarity indexes.

## 5. Conclusions

Future prospective studies are needed in order to determine whether the predominant polarity is indeed one clinical factor that might guide the clinician in choosing the right mood stabilizer (e.g., the one with a PI > 1.0) for the right patient (e.g., the one with MPP), at the right stage of illness (after several cycles of the disorder).

Our study also emphasizes the need of more real-world studies aimed at determining which sociodemographic and clinical characteristics may help clinicians in personalizing maintenance treatments for BD, in order to achieve higher response rates, which is the ultimate goal of the clinical characterization of the disorder.

## Figures and Tables

**Table 1 medicina-57-00598-t001:** Socio-demographic and clinical characteristics of bipolar disorder patients (*n* = 653).

Variable	
Female, *n* (%)	388 (59.4)
Age at interview (years), mean (SD)	50.6 (15.6)
Age at onset (years), mean (SD)	30.3 (12.9)
Illness duration (years), mean (SD)	20.2 (13.9)
Employment ^¥^	
Employed, *n* (%)	119 (18.3)
Unemployed, *n* (%)	368 (56.4)
Retired, *n* (%)	165 (25.3)
Marital status, *n* (%)	
Single	186 (28.5)
Married/Cohabiting	341 (52.2)
Divorced	84 (12.9)
Widowed	42 (6.2)
Diagnosis, *n* (%)	
Bipolar disorder type I	262 (40.1)
Bipolar disorder type II	371 (56.8)
Unspecified bipolar disorder	19 (2.9)
Substance/medication induced bipolar disorder	1 (0.2)
Polarity of first episode, *n* (%)	
Hypo/manic	211 (32.3)
Depressive	418 (64.0)
NA	24 (3.7)
Predominant polarity_2/3_, *n* (%)	
Depressive	236 (36.1)
Hypo/manic	70 (10.7)
None	347 (53.1)
Predominant polarity_50%_, *n* (%)	
Depressive	361 (55.3)
Hypo/manic	118 (18.1)
None	174 (26.6)
Prescribed mood stabilizers/antipsychotics, *n* (%)	
Lithium	407 (62.3)
Valproate	209 (32.0)
Carbamazepine	14 (2.1)
Oxcarbazepine	8 (1.2)
Lamotrigine	52 (7.9)
Olanzapine	81 (12.4)
Quetiapine	155 (23.7)
Clozapine	7 (1.1)
Risperidone	24 (3.7)
Paliperidone	3 (0.5)
Ziprasidone	4 (0.6)
Asenapine	28 (4.3)
Amisulpride	8 (1.2)
Haloperidol	27 (4.1)
On antidepressants, *n* (%)	272 (41.7)

SD—Standard Deviation; *p*—*p* value; ^¥^ 1 missing data.

**Table 2 medicina-57-00598-t002:** Comparison of clinical correlates between manic predominant polarity and depressive predominant polarity bipolar disorder patients.

**Clinical Variable**	**Manic Predominant Polarity_2/3_ (*n* = 70)**	**Depressive Predominant Polarity_2/3_ (*n* = 236)**	**χ*^2^* or *t*/U**	***p***
Female, *n* (%)	33 (47.1)	144 (61.0)	4.2	**0.05**
Age at interview (years), mean (SD)	44.4 (15.4)	52.8 (15.0)	−4.1	**<0.0001**
Age at onset (years), mean (SD)	29.5 (13.65)	31.9 (13.6)	−1.3	0.2
Presence of family history of any psychiatric disorder, *n* (%)	47 (67.1)	144 (61.0)	0.9	0.4
Presence of family history of mood disorder, *n* (%)	41 (58.6)	127 (53.8)	0.5	0.5
Presence of family history of bipolar disorder, *n* (%)	14 (20.0)	43 (18.2)	0.11	0.73
Number of first- and second-degree family members affected by psychiatric disorders, mean (SD)	0.77 (0.8)	0.76 (0.9)	0.07	0.9
Number of manic episodes, mean (SD)	2.9 (3.1)	0.4 (0.9)	3536.0	**<0.0001**
Number of hypomanic episodes, mean (SD)	2.6 (3.4)	1.5 (1.5)	8207.0	0.9
Number of depressive episodes, mean (SD)	1.5 (1.5)	6.0 (4.4)	1413.0	**<0.0001**
Total number of episodes, mean (SD)	7.3 (5.3)	8.3 (5.6)	−1.3	0.2
Illness duration (years), mean (SD)	14.85 (12.5)	20.9 (13.8)	−1.3	0.2
Number of hospital admissions, mean (SD)	2.5 (0.8)	2.4 (1.8)	0.18	0.8
Presence of lifetime suicidal behavior, *n* (%)	5 (7.1)	58 (24.7)	10.1	**0.001**
Type of clinical course cycle				
MDI, *n* (%)	10 (14.3)	28 (11.9)	4.5	0.3 ^^^
DMI, *n* (%)	4 (5.7)	34 (14.4)		
Irregular cycling, *n* (%)	55 (78.6)	169 (71.6		
Continuous cycling, *n* (%)	1 (1.4)	3 (1.3)		
Rapid cycling, *n* (%)	0 (0.0)	2 (0.8)		
**Clinical Variable**	**Manic Predominant Polarity_50%_ (*n* = 118)**	**Depressive Predominant Polarity_50%_ (*n* = 236)**	***χ^2^* or *t*/U**	***p***
Female, *n* (%)	61 (51.7)	233 (64.5)	6.2	**0.016**
Age at interview (years), mean (SD)	46.4 (15.8)	52.9 (14.6)	−4.1	**<0.0001**
Age at onset (years), mean (SD)	29.1 (13.3)	30.7 (12.9)	−1.2	0.23
Presence of family history of any psychiatric disorder, *n* (%)	82 (69.5)	234 (64.8)	0.9	0.37
Presence of family history of mood disorder, *n* (%)	74 (62.7)	211 (58.4)	0.7	0.45
Presence of family history of bipolar disorder, *n* (%)	31 (26.3)	70 (19.4)	2.5	0.12
Number of first- and second-degree family members affected by psychiatric disorders, mean (SD)	0.8 (0.8)	0.8 (0.9)	0.1	0.8
Number of manic episodes, mean (SD)	2.8 (3.1)	0.6 (1.3)	9882.0	**<0.0001**
Number of hypomanic episodes, mean (SD)	2.9 (3.4)	2.3 (2.6)	21,103.0	0.87
Number of depressive episodes, mean (SD)	2.7 (2.5)	6.2 (4.2)	7898.0	**<0.0001**
Total number of episodes, mean (SD)	8.8 (6.0)	9.5 (6.5)	−1.0	0.3
Illness duration (years), mean (SD)	17.1 (13.2)	22.1 (13.4)	−3.6	**<0.0001**
Number of hospital admissions, mean (SD)	2.7 (0.9)	2.7 (2.2)	126.0	0.34
Presence of lifetime suicidal behavior, *n* (%)	19 (16.1)	92 (25.6)	4.5	**0.034**
Type of clinical course cycle				
MDI, *n* (%)	39 (33.1)	77 (21.3)	11.9	**0.018** ^^^
DMI, *n* (%)	9 (7.6)	58 (16.1)		
Irregular cycling, *n* (%)	69 (58.5)	215 (59.6)		
Continuous cycling, *n* (%)	1 (0.8)	4 (1.1)		
Rapid cycling, *n* (%)	0 (0.0)	7 (1.9)		

MDI*—*(hypo)mania—depression—free interval; DMI—depression—(hypo)mania—free interval; SD—Standard Deviation; *p*—*p* value; ^^^—χ^2^ bootstrap with 1000 samples.

**Table 3 medicina-57-00598-t003:** Logistic regression of clinical characteristics associated with predominant polarity subgroup defined according to the 50% or 2/3 of prevalent episodes criterion.

Outcome (Dependent Variable)	Independent Variable	β	SE	OR	95% CI Lower	95% CI Upper	*p*
Depressive predominant polarity_2/3_	Age at interview	0.04	0.01	1.04	1.02	1.06	<0.0001
	Presence of lifetime suicidal behavior	−1.54	0.5	0.2	0.08	0.6	0.002
Depressive predominant polarity_50%_	Age at interview	0.02	0.009	1.02	1.001	1.04	0.035
	Presence of lifetime suicidal behavior	−0.6	0.29	0.5	0.3	0.96	0.035

OR—odds ratio; SE—standard error; CI—confidence interval; *p*—*p*-value.

**Table 4 medicina-57-00598-t004:** Comparison of polarity index values among predominant polarity subgroups defined according to the 50% or 2/3 of prevalent episodes criterion.

**Polarity Index**	**Manic Predominant Polarity_2/3_ (*n* = 16)**	**Depressive Predominant Polarity_2/3_ (*n* = 148)**	**No Predominant Polarity_2/3_ (*n* = 186)**	**F**	***p***
Polarity index (Popovic), mean (SD)	1.84 (0.57)	1.56 (0.83)	1.61 (0.62)	1.18	0.31
Polarity index (Carvalho), mean (SD), mean rank	1.37 (1.18)	1.52 (1.59) ^$^	1.57 (1.24)	0.16	0.85
**Polarity Index**	**Manic Predominant Polarity_50%_ (*n* = 36)**	**Depressive Predominant Polarity_50%_ (*n* = 218)**	**No Predominant Polarity_50%_ (*n* = 96)**	**F or χ^2^**	***p***
Polarity index (Popovic), mean (SD)	1.91 (0.77)	1.54 (0.76)	1.61 (0.57)	4.038	**0.018**
Polarity index (Carvalho), mean (SD), mean rank	1.79 (1.94), 170.89	1.49 (1.41), 169.69 ^$^	1.56 (1.09), 188.55	2.68	0.26

SD—Standard Deviation; *p*—*p* value; Significant values are typed in bold. ^$^ 1 missing data.

## Data Availability

The data that support the findings of this study are not openly available, due to confidentiality of data collected from hospital clinical charts.

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
