# Peer review of "Predominant Polarity and Polarity Index of Maintenance Treatments for Bipolar Disorder: A Validation Study in a Large Naturalistic Sample in Italy"

_medicina, 2021, doi:10.3390/medicina57060598_

Round 1

Reviewer 1 Report

This is an important study given the challenges to diagnosing and properly treating bipolar disorder (BD). There are some grammatical errors in the paper (which may be due to language issues), that need to be addressed. At places in the paper, it is difficult to adequately understand what the authors are conveying (rephrasing those sections would be helpful). That said, they do a nice job outlining the issue of predominant polarity and the potential usefulness it may have in treating patients the BD. Although not all hypotheses were supported, the authors do report interesting clinical findings related to polarity. Below are specific points for consideration:

  • Line 40, should be which in turnÂť
  • Line 68 (and in other places), Authors is not capitalized.
  • Lines 81-83, the authors note age at interview. Do they mean age at first diagnosis? This should be clarified.
  • Line 94, a clearer discussion of the polarity index is needed. It would help readers to know how exactly the value is calculated (provide more detail).
  • Lines 103-109 are confusing, please restate. What is meant by a mood stabilizer with a higher PI.? Perhaps higher PI based on medication status?Âť
  • Line 169, more detail needs to be provided on how the polarity index was calculated.
  • Line 176, the authors note that predominant polarity was defined as at least half (PP50%) or two-thirds (PP2/3) of a 176 patients past episodes fulfilling DSM-5 criteria for major depressive episode or manic or 177 hypomanic episode. This would appear that by default, the 50% group would be subsumed into the 2/3 group. Clarification of this would be good for readers. Although it would seem obvious, a simple statement to clarify, and justification for examining both types of criteria would be helpful.
  • What were the medications participants were taking? I don't see a description of this? Perhaps individual differences may have influenced the findings?
  • A more detailed explanation for the lack of significant findings and a description of future investigations in this area is needed on the Discussion section.

Author Response

Comments from the Editor in Chief:

  1. Please indicate when the study was carried out.

Response: We added in the Materials and Methods the following sentence in order to clarify this issue:The retrospective examination of clinical charts of patients was carried out from June 2018 to December 2019. Statistical analyses concerning the present study were performed in December 2020.”

  1. Table 1 could be moved to the results section as it provides the numbers and percentages calculated.

Response: we moved this table to the results section as suggested.

  1. Please add 95% Cis for ORs in Table 3.

Response: we apologize for not having provided CIs. The new Table 3 includes 95% CIs for ORs.

Reviewer Number 1

This is an important study given the challenges to diagnosing and properly treating bipolar disorder (BD). There are some grammatical errors in the paper (which may be due to language issues), that need to be addressed. At places in the paper, it is difficult to adequately understand what the authors are conveying (rephrasing those sections would be helpful). That said, they do a nice job outlining the issue of predominant polarity and the potential usefulness it may have in treating patients the BD. Although not all hypotheses were supported, the authors do report interesting clinical findings related to polarity.

Response: We thank reviewer number 1 for his/her comments. We tried to address grammatical errors, as suggested, and do apologize for this.

Below are specific points for consideration:

  • Line 40, should be which in turn

Response: we corrected the error.

  • Line 68 (and in other places), Authors is not capitalized.

Response: we corrected here and in other places.

  • Lines 81-83, the authors note age at interview. Do they mean age at first diagnosis? This should be clarified.

Response: in the few studies which examined the issue, it was age at interview and not age at first diagnosis. Being confusing and not so relevant, we decided to delete the note referring to age at interview.

  • Line 94, a clearer discussion of the polarity index is needed. It would help readers to know how exactly the value is calculated (provide more detail).

Response: we extended the paragraph, adding how exactly Popovic and colleagues calculated the PI: “Polarity index was firstly retrieved by Popovic and colleagues (2012): they systematically reviewed randomized controlled trials of medications used for the maintenance treatment of BD (mood stabilizer or antipsychotic drug alone or in combination with a mood stabilizing agent such as lithium or valproate versus placebo). Number Needed to Treat (NNT) was calculated by taking the reciprocals of the differences between the rates of the outcomes for two interventions; the polarity index was then retrieved by dividing NNT for prevention of depressive episodes and NNT for prevention of manic episodes. A PI value above 1.0 indicates a relative greater antimanic prophylactic efficacy while number below 1.0 a relative greater antidepressive efficacy [39].”

  • Lines 103-109 are confusing, please restate. What is meant by a mood stabilizer with a higher PI.? Perhaps higher PI based on medication status?

Response: we are sorry for being confusing. We tried to be more clear and restated the paragraph as follows: “In recent years, few authors investigated whether the choice of the right mood stabilizer over the long-term in terms of PI is influenced by the predominant polarity as a course specifier of the disorder (that is whether clinicians prescribe more often to individuals with a manic predominant polarity a mood stabilizer with greater effectiveness in preventing manic episodes over depressive ones and thus with a higher PI). No prospective, longitudinal studies exist examining the relative effectiveness of a mood stabilizer chosen for its polarity index according to a clinical characterization based on the predominant polarity. We have only few retrospective studies which evaluated cross-sectionally whether subjects with MPP received indeed mood stabilizers with a higher PI (that is with greater effectiveness in preventing manic episodes over depressive ones) as compared to subjects with DPP.“

  • Line 169, more detail needs to be provided on how the polarity index was calculated.

Response: we extended this methods section including the following sentences:

Polarity index was calculated for each patient according to the method described in Popovic et al. (2012) [39](Polarity Index Popovic) and that in Carvalho et al. (2015) [40](Polarity Index Carvalho). Popovic and colleagues [39] included 16 maintenance trials, while Carvalho et al. [40] 18. NNT was calculated by taking the reciprocals of the differences between the rates of the outcomes between the interventions; the PI was retrieved by dividing the NNT for prevention of depressive episodes by NNT for prevention of manic ones.

According to Popovic at al. [39], the PI values for each drug were as follows: 12.09 for risperidone LAI, 4.38 for aripiprazole, 3.91 for ziprasidone, 2.98 for olanzapine, 1.39 for lithium, 1.14 for quetiapine, 0.62 for oxcarbazepine, 0.49 for valproate and 0.40 for lamotrigine. Carvalho and colleagues [40] recalculated, on the basis of different and new trials, the polarity indexes of maintenance drugs for BD and provided the following PI for each compound: 10.4 for aripiprazole monotherapy, 9.1 for risperidone LAI, 4.2 for aripiprazole adjunctive to lithium/divalproex, 4.0 for olanzapine monotherapy, 3.9 for ziprasidone adjunctive to lithium/divalproex, 1.4 for lithium and for quetiapine monotherapy, 0.8 for quetiapine combined with lithium/divalproex, 0.6 for oxcarbazepine combined with lithium, 0.5 for olanzapine combined with lithium/divalproex and for divalproex, and 0.4 for lamotrigine.

Although a PI was assigned to valproate and oxcarbazepine, it was considered unreliable because the pivotal trials did not show statistical superiority of those two drugs over placebo. We then excluded from the analysis patients prescribed drugs without an assigned PI (e.g. patients on valproate or oxcarbazepine, or on antipsychotic such as paliperidone).”

  • Line 176, the authors note that predominant polarity was defined as at least half (PP50%) or two-thirds (PP2/3) of a 176 patients past episodes fulfilling DSM-5 criteria for major depressive episode or manic or 177 hypomanic episode. This would appear that by default, the 50% group would be subsumed into the 2/3 group. Clarification of this would be good for readers. Although it would seem obvious, a simple statement to clarify, and justification for examining both types of criteria would be helpful.

Response: we clarified with a sentence that the PP50% group would be subsumed into the PP2/3 one. Moreover, we added a brief justification for examining both types of criteria: “By default, the PP50% group would be subsumed into the PP2/3 group. We chose to consider also the 50% predominant polarity criterion as some authors previously suggested to be less stringent in defining this course specifier, which is easier to be used in clinical practice [21-26].”

  • What were the medications participants were taking? I don't see a description of this? Perhaps individual differences may have influenced the findings?

Response: we added in table 1 (results section) the N (%) of subjects taking each mood stabilizers (see also comments from reviewer 2).

  • A more detailed explanation for the lack of significant findings and a description of future investigations in this area is needed on the Discussion section.

Response: we extended the discussion section as follows: “The lack of significant findings may have several possible explanations: the first one is that Italian psychiatrists may prescribe maintenance treatments without considering the predominant polarity of individuals with BD, and do not consider in the real-world the PI of a compound to be a reliable factor that may guide the choice of a single prophylactic treatment. Again, psychiatrists in Barcelona may be more sensitive to the PP and PI concepts when choosing a maintenance treatment for their patients being the PI proposed for the first time by the group of the Barcelona Bipolar Disorder Program. Another possible explanation of the lack of significant findings is that we did not control for the length of previous exposure to maintenance treatments; it is possible that people with more stable maintenance treatments (prophylactic treatment maintained stable for months/years) could show differences in the mean PI of their treatments according to the PP. Another issue to be considered is that we assigned a PI to each drug without considering whether it was used at an appropriate dosage.

Given that the methodology used has several limitations (the cross-sectional evaluation of mood stabilizers prescribed does not take into account clinical factors other than the PP that might have guided the choice of the compound for the individual patient, nor we know and controlled for the length of exposure to the mood stabilizer prescribed), we cannot conclude that the PI construct is not useful in clinical practice.

Prospective, longitudinal studies could in the near future examine whether prescribing mood stabilizers with a higher PI (thus a greater efficacy in preventing manic episodes over depressive ones) to patients with a MPP course of the disorder (and vice versa prescribing compounds with lower PI to individuals with DPP) is associated with greater effectiveness in stabilizing patients. Several socio-demographic and clinical factors other than the PP contribute to the choice of the prescribed stabilizing drug, so that future studies will help clinicians to identify those factors which may be more informative and reliable in predicting response to mood stabilizers over the long-term.

More randomized controlled trials on old, off-patent, but still widely used drug such as valproate and carbamazepine are also needed, in order to help predict/refine their polarity indexes.”

Reviewer 2 Report

The manuscript is well-written and provides another validation study on an important topic that has previously generated mixed results.

The Introduction is informative. The citation number 18 of Angst and colleagues who proposed the concept of predominant polarity, does raise the issue of bipolar spectrum disorders and the controversy over putative milder/briefer mood lability that may or may not represent milder versions of BD-I versus personality disorders and adjustment/stress reactions.  Does the definition of bipolar disorder itself impact on the wide variation in categorizing predominant polarity as noted in the paragraph covering lines 73-78?  This issue of diagnostic specificity could be mentioned in the Introduction as a possible reason for such variation if those references cited in lines 73-78 represent different cut-offs on the ‘bipolar spectrum’.

It also raises the question whether a clearer difference might emerge between MPP and DPP or noPP – if only BD-I cases are considered?  Could the authors run an analysis on the BD-I subset of their sample?

A simple table that listed medications according to polarity index would be useful for readers, who may not have time to look to Popovic and Carvalho’s articles.  The fact that valproate and carbamazepine do not have sufficient data to predict their polarity index is a significant limitation – this manuscript would be enhanced if data of medications used by this Italian cohort were included perhaps in a table in the Results section, and the numbers on valproate and carbamazepine could be seen - however divalproex is listed with a PI in the articles by  Popovic and Carvalho. 

The call for prospective studies is appropriate in the Conclusion but could be added to by a call for more RCTs on old, off-patent, but still widely used valproate and carbamazepine that could help predict/refine their polarity indexes.

Author Response

Reviewer 2

The manuscript is well-written and provides another validation study on an important topic that has previously generated mixed results.

Response: We thank reviewer number 2 for his/her comments.

The Introduction is informative. The citation number 18 of Angst and colleagues who proposed the concept of predominant polarity, does raise the issue of bipolar spectrum disorders and the controversy over putative milder/briefer mood lability that may or may not represent milder versions of BD-I versus personality disorders and adjustment/stress reactions.  Does the definition of bipolar disorder itself impact on the wide variation in categorizing predominant polarity as noted in the paragraph covering lines 73-78?  This issue of diagnostic specificity could be mentioned in the Introduction as a possible reason for such variation if those references cited in lines 73-78 represent different cut-offs on the ‘bipolar spectrum’.

It also raises the question whether a clearer difference might emerge between MPP and DPP or noPP – if only BD-I cases are considered?  Could the authors run an analysis on the BD-I subset of their sample?

Response: we agree with reviewer 2 comments. It is possible that different approaches to the diagnosis of BD (e.g. tendencies to diagnose personality disorders instead of BD, or failing to recognize hypomanic episodes in the natural course of the disorder thus diagnosing recurrent MDD instead of BD) could have contributed to selecting different groups of individuals with BD. We added a brief comment in the Introduction: “This wide variation may be partly explained by differences in the samples included in each study, for example in the proportion of patients with BD type I versus type II. Moreover, another contributor to the wide variation found between different studies may be the definition of bipolar disorder itself; it is possible that different approaches to the diagnosis of BD (e.g. tendencies to overdiagnose personality disorders instead of milder forms of BD, or failing to recognize hypomanic episodes in the history of patients with recurring depressive episodes thus resulting in a diagnosis of recurrent MDD instead of BD type II) could have contributed to selecting different groups of individuals with BD in each study.”

Moreover, we run an analysis on the BD-I sample, and did not find other significant differences in the mean PI. We added a sentence in the result section, as follows: “When we run the analysis on the subset of patients with BD type I we did not find other significant differences.”

A simple table that listed medications according to polarity index would be useful for readers, who may not have time to look to Popovic and Carvalho’s articles.  The fact that valproate and carbamazepine do not have sufficient data to predict their polarity index is a significant limitation – this manuscript would be enhanced if data of medications used by this Italian cohort were included perhaps in a table in the Results section, and the numbers on valproate and carbamazepine could be seen - however divalproex is listed with a PI in the articles by  Popovic and Carvalho. 

Response: considering also reviewer 1 suggestions, we included in the method section a paragraph with polarity indexes calculated by Popovic and Carvalho, also mentioning differences in the way they calculated PIs:

 Polarity index was calculated for each patient according to the method described in Popovic et al. (2012) [39](Polarity Index Popovic) and that in Carvalho et al. (2015) [40](Polarity Index Carvalho). Popovic and colleagues [39] included 16 maintenance trials, while Carvalho et al. [40] 18. NNT was calculated by taking the reciprocals of the differences between the rates of the outcomes between the interventions; the PI was retrieved by dividing the NNT for prevention of depressive episodes by NNT for prevention of manic ones. According to Popovic at al. [39], the PI values for each drug were as follows: 12.09 for risperidone LAI, 4.38 for aripiprazole, 3.91 for ziprasidone, 2.98 for olanzapine, 1.39 for lithium, 1.14 for quetiapine, 0.62 for oxcarbazepine, 0.49 for valproate and 0.40 for lamotrigine. Carvalho and colleagues [40] recalculated, on the basis of different and new trials, the polarity indexes of maintenance drugs for BD and provided the following PI for each compound: 10.4 for aripiprazole monotherapy, 9.1 for risperidone LAI, 4.2 for aripiprazole adjunctive to lithium/divalproex, 4.0 for olanzapine monotherapy, 3.9 for ziprasidone adjunctive to lithium/divalproex, 1.4 for lithium and for quetiapine monotherapy, 0.8 for quetiapine combined with lithium/divalproex, 0.6 for oxcarbazepine combined with lithium, 0.5 for olanzapine combined with lithium/divalproex and for divalproex, and 0.4 for lamotrigine. Although a PI was assigned to valproate and oxcarbazepine, it was considered unreliable because the pivotal trials did not show statistical superiority of those two drugs over placebo. We then excluded from the analysis patients prescribed drugs without an assigned PI (e.g. patients on valproate or oxcarbazepine, or on antipsychotic such as paliperidone).”

We also specified in the introduction that Popovic et al. 2012 provided a PI for valproate although they considered it unreliable as in the pivotal trial valproate did not separated from placebo in preventing relapses. That’s why in the following validation studies by Popovic et al. 2014 and Volkert et al. 2014 they did not include the PI for valproate. We added the following sentence: “A limitation of these studies is that they did not include valproate and carbamazepine in their analyses, as a PI to carbamazepine was not assigned by Popovic and colleagues [39] due to the lack of a long-term placebo-controlled trial, and the PI for valproate was considered unreliable because the pivotal trial did not show statistical superiority of valproate over placebo. Moreover, PI for newer agents such as paliperidone or asenapine were lacking.”

Moreover, we added in table 1 – results – the N (%) of individuals taking each mood stabilizer (including N taking valproate or carbamazepine).

The call for prospective studies is appropriate in the Conclusion but could be added to by a call for more RCTs on old, off-patent, but still widely used valproate and carbamazepine that could help predict/refine their polarity indexes.

Response: thank you for the suggestion. According to reviewer 1 suggestions we expanded the discussion and according to your suggestion we included a call for more RCTs on valproate or carbamazepine. At the end of the discussion we added this sentence: “More randomized controlled trials on old, off-patent, but still widely used drug such as valproate and carbamazepine are also needed, in order to help predict/refine their polarity indexes.”